# Human–Animal Interactions: Expressions of Wellbeing through a “Nature Language”

**DOI:** 10.3390/ani11040950

**Published:** 2021-03-29

**Authors:** Rachel M. Yerbury, Samantha J. Lukey

**Affiliations:** 1llawarra Health and Medical Research Institute, University of Wollongong, Wollongong, NSW 2522, Australia; 2School of Environment, Engineering and Science, Southern Cross University, Lismore, NSW 2480, Australia; 3School of Health & Society, Faculty of the Arts, Social Sciences and the Humanities, University of Wollongong, Wollongong, NSW 2522, Australia; slukey@uow.edu.au

**Keywords:** human–animal interactions, wildlife, nature interactions, human wellbeing, reciprocity

## Abstract

**Simple Summary:**

This article is an exploration of human–animal interactions (HAIs) in nature (in the wild). It explores how animal encounters may have positive mental health benefits for people and also considers the potential role of humans giving back to nature and wildlife. This idea is examined in terms of people’s connectedness to nature through an Interaction Pattern approach. The Interaction Pattern approach provides a means through which people can express the meaningful ways they interact with animals and nature and the psychological outcomes that result. This study listens to the responses of more than three hundred people who explain how an encounter with a wild marine animal affected their connection to nature. The findings suggest that when people encounter and interact with animals in their natural environment, their mental health and wellbeing is enhanced; feelings of love, belonging, positive feelings, fulfillment and the gaining of perspective, are articulated. It is suggested that human wellbeing from HAIs needs to include the wellbeing and perspective of nature and animals to promote two-way benefits.

**Abstract:**

Human–animal interactions (HAIs) can be beneficial for humans in a number of ways, and interactions with wild animals may contribute to human mental wellbeing, partly through nature connectedness. This study applies the “Nature Interaction Pattern” approach (proposed by Kahn and colleagues) to characterize the structure of meaningful human engagement with nature and animals, and to consider the wellbeing outcomes. This qualitative, retrospective study uses open responses from 359 participants who describe how their wild animal encounters affected their nature connectedness. Thematic analysis explores five nature Interaction Patterns and four resulting Psychological Descriptions that occur in the portrayals of the marine animal encounters and these are described using representative quotes. Feelings of love, belonging, positive feelings, fulfillment and the gaining of perspective, were linked with the human–animal experience and the Interaction Patterns. These findings suggest that when people encounter and interact with animals in their natural environment, their mental health and wellbeing may be enhanced. Further, through connecting with nature and animals, reciprocity may occur, that is, when people connect with nature and animals, they can also give back. Within this interaction there becomes an entanglement of experiences thereby encouraging caring for nature and animals.

## 1. Introduction

### 1.1. Human and Animal Interactions

Human lives intersect with non-human animals in myriad ways, thus human and animal interactions (HAIs) are broad and varied. Animals form an integral part of human life: from companion animals in the home, to the cattle we graze to eat, to insects and birds in our backyards, to the wildlife we encounter in the wild or in zoos. In some First Nations worldviews, HAIs are informed by a spiritual connection to all sentient beings where animals are considered sacred, equal and possessing of healing powers [1,2,3]. The act of ongoing colonial hijacking and attempted erasure of First Nations knowledges has resulted in reduced understanding of human-nature interactions [3]. Therefore, many of the spiritual and symbiotic relationships between indigenous humans and animals (for example, mutually beneficial hunting) have been overlooked through the process of colonization [4] (The authors acknowledge that there are multiple terms used to describe the First Nations people of Australia. For the purposes of this paper “indigenous” will be used when discussing international indigenous perspectives and “Indigenous” will be used when discussing Australian Aboriginal and Torres Strait Islander people.). In order to understand how HAIs can benefit both human wellbeing and contribute to earth wellbeing, it is therefore crucial to include indigenous voices in this discussion. Through a decolonizing lens, what has been lost or disregarded in the history of HAIs can be acknowledged [3,5]. Today the roles that animals enact within various human contexts, systems and cultures are diverse, but are generally less naturalistic and predominantly utilitarian [6]. Different human viewpoints and attitudes inform the way animals are treated and valued in HAIs [7]. Animals are rarely considered equal and are undervalued in terms of intrinsic value as they are predominantly seen through an anthropocentric lens. Hence, there has been a call for a re-thinking of the utilitarian view [6] with the possibility of returning to the a more ecocentric indigenous worldview in conceptualizing HAIs [3,4].

### 1.2. Animals and Human Needs

The World Health organization constitution states that human mental health is “a state of complete physical, mental, and social wellbeing” [8]. Some of the signifiers of wellbeing include positive emotions [9], belonging [10], love and meaningful relationships [11]. Wellbeing also includes the motivation to achieve personal growth and fulfilment [12], and human wellbeing profits through seeing beyond the self to the bigger picture and incorporating an ecocentric worldview [13].

It is possible that interactions with animals can perform multiple roles in satisfying human needs [14]. Animals may assist humans to meet basic physiological and safety needs (agriculture, shelter, transport, protection and resource collection) as well as higher order psychological needs (love and belonging, esteem and fulfillment) [15]. Further, it is suggested that HAIs could support human need fulfillment by connecting people with the natural world “what we yearn for, then, is to be on more familiar terms with other beings, to be a part of it all” [5].

The human psychological benefits of human–animal interactions (HAI) are well documented. Some studies focus on how domesticated animals and companion animals reduce stress, anxiety and loneliness [14,16,17]. However, the results regarding companion animals and human mental health are far from conclusive [18]. An emerging area of HAI research is the human mental health benefits of interactions between humans and wildlife [13,19,20,21,22,23]. As has been discussed elsewhere [20] interactions with wildlife can benefit human wellbeing by contributing to positive emotions, engagement, relationships and connection, meaning, achievement and also an ecocentric perspective [13]. In particular, moving towards reciprocal benefits as evidenced by environmental behaviors may be enhanced by marine wildlife experiences [24].

### 1.3. Animals and Nature Connectedness

In examining human benefits from wild animal interactions, studies have considered the benefits of connecting with nature as part of the animal experience [23,25,26]. The first author has reported elsewhere [27] that nature connectedness mediates the relationships between contact with animals and wellbeing. This conclusion is based on the concept that there is an innate human need for people to feel they have bonded with nature and animals (as components of nature), and that humans are not whole without this relationship [28,29]. As Shepherd maintains, “The human species emerged enacting, dreaming and thinking animals and cannot be fully itself without them” [29] (p. 4). Following on from connecting with nature and animals is the vision of reciprocity; when people join with nature and animals, they can also give back by caring for the land and honoring the animals they take from as do the Maori cultures [30]. A sense of mutuality in experiences encourages caring for and conserving nature and animals [24,26,31]. 

### 1.4. Human–Animal Encounters Expressed as Nature Interaction Patterns

To examine the effects of HAIs for humans, we need to understand the way that people respond to and interpret and articulate interactions. Moreover, further exploration is needed to understand what this means to them and the implications this has in caring for nature. Yunkaporta [4] explains from an Indigenous perspective “the stories are passed down and people partner with whales and dolphins and others to continue caring for the country beneath the sea” (p. 79). To pass on such ideas and narratives, a community needs a shared language to communicate the meanings and knowledge that humans have learnt about nature and animals. When contact with the natural world is reduced or lost, the stories and even the language to describe nature interactions are also lost [5]. If the language of nature interactions becomes unfamiliar and indefinable, the culture of interaction and kinship with other beings is fragmented [32]. As Fisher [5] articulates, “the meanings we find in relation to nature can never be other than what our existing language-forms already say” (p. 64). To revive the culture of nature interaction and connectedness, Kahn and colleagues have proposed ‘a Nature Language’ [33] as a dialect for expressing how people interact with nature. Within a Nature Language, Interaction Patterns (IPs) help to characterize the structure of meaningful human engagement with nature [33]. Keystone Interaction Patterns refer to the most meaningful, foundational and frequently occurring Interaction Patterns [34]. Because these patterns are meaningful and holistic, they are fused with emotion and integrate psychological experiences [33]. Psychological Descriptions (PDs) are used in an Interaction Pattern approach to encompass cognitive and emotional reflections [33], and are defined as “the portion of the participant’s Nature Language that describes their personal reflections on and feelings about their experience” [35]. In order to provide a deeper understanding of how HAIs affect human wellbeing through a nature connectedness lens, this study applies the nature interaction pattern approach of Kahn and colleagues, using keystone interactions patterns to understand the themes [33,35,36].

### 1.5. Aim and Research Question

This qualitative, retrospective study explores how human–wildlife interactions in nature can contribute to human mental health.

The study aims to examine the potentiality of an animal-specific adaptation of the nature language and interaction approach to conceptualize and interpret human–animal relationships and benefits [33,35,36]. The proposition of this study is that human mental health benefits occur within the nature connection of enacting Interaction Patterns with wild animals.

The specific research question is: Can Interaction Patterns provide an understanding of the human mental health benefits of human–animal interactions in nature? To examine this question, the study explores both nature Interaction Patterns and Psychological Descriptions that occur in portrayals of marine animal encounters.

## 2. Materials and Methods

### 2.1. Ethics Approval

The collection of the data was conducted under the approval of the Southern Cross University Human Ethics Committee (ECN-15-136), while the first author was a PhD candidate.

### 2.2. Recruitment and Data Collection

Participants were recruited for the study via social media posts (Facebook and Twitter) and also via three Australian wildlife ecotour company’s customer emails. The online survey (www.qualtrics.com) included a question asking participants to explain how their experiences watching or swimming with marine wildlife (specifically marine mammals) affected their connection to nature, which provided the qualitative data for the study (a subset of data from a larger study [13]). Some of the participants did describe marine mammal encounters but many took the question more broadly to encompass any marine animal. Therefore, this open response question, while being uniform and focused [37], allowed scope for the expression of meanings, personal interpretations and viewpoints that were examined individually and collectively [37]. This approach is supported by Yunkaporta [4] who highlights that we have to compare our stories with the stories of others to seek greater understanding about our reality” (p. 129). 

### 2.3. Participants

The majority of the 359 participants were Australian (96%) and female (77%). Age was evenly spread, with 17–23% of participants in each of five age categories from 13–25 years of age to 56 years and over. 

### 2.4. Data Analysis and Reliability

Data was exported into NVivo 12 (Mac) and the first author read through the responses to generate initial thoughts about participant expressions of animal interactions. This research was conducted within a realist/essentialist approach in order to convey the subjective meanings and realities of participants. To examine the research question, exploration of the Interaction Patterns, Psychological Descriptions and their overlaps used thematic analysis in Nvivo 12 (Mac). The researchers were an active part of the interpretation of the data, through their positionality and values [38]. Theoretical or “top down” thematic analysis at the latent level was employed to identify and report patterns through Braun and Clarke’s six-steps [38]. Whilst this paper is predominantly qualitative, numbers of responses to particular categories have been included. 

The first author coded the responses to generate initial thoughts about participant expressions of animal interactions. Themes of Interaction Patterns as well as emergent Psychological Descriptions, were guided by the research of Kahn and colleagues [34,35]. The first author sought guidance from Peter Kahn and Elizabeth Lev [39] in using the IP approach, and re-worked the coding process to ensure that Interaction Patterns were coded separately from the linked Psychological Descriptions and adequate coding guidelines were created.

The qualitative responses were read and re-read to enable familiarity, and initial codes of interest were manually scribed and then mapped to identify patterns across the data [38]. Five previously identified Keystone Interaction Patterns [33,36] were deemed to encompass the breadth of the dataset from the consideration of responses. As the coding of IPs was undertaken, four Psychological Descriptions (PDs) were identified from the responses (refer Table 1 for IP and Table 2 for PD themes and definitions). While not attempting to be comprehensive in the evaluation of mental health, the four Psychological Descriptions were used as indicators of mental wellbeing for the purposes of this study. Representative verbatim quotes are included, to illustrate themes [40,41].

Reliability and rigor were pursued by a verification step (reviewing for discrepancies and errors) by the original analyst (first author) and credibility checks by an “additional analytic auditor” (second author) [41] (p. 222). The auditor initially coded 5% of the data as part of the iterative process of clarifying and revising the themes and generating the codebook with definitions, decision rules and examples (refer Appendix A). As is described more fully in the Codebook, sections of each response were coded according to IPs and PDs. While the IP coding only included the interaction patterns, the Psychological Description also encompassed contextual text as recommended in the literature [38]. The responses were only coded to one IP but could be coded to more than one PD if necessary, as described in the codebook (Appendix A). Once the codebook was finalised, the auditor randomly coded 10% of the data as guided by previous research [42]. Further iterations through thorough discussions and negotiated agreement between authors occurred [43] until acceptability was determined, and the codes were re-applied to the whole data set. The agreement percentages ranged between 81.34% to 98.78%. Intercoder agreement can be preferable over other reliability measures when the intercoder agreement is very high. [42,44]. According to Cresswell and Poth (2016) naturalistic research seeks “dependability rather than reliability… and confirmability rather than objectivity” [45] (p. 204). These validation methods have been ensured in this study by the inclusion of the analytic auditor of the research process [37] and have been used in other studies to guard against bias and to ensure verifiability [46].

## 3. Results

### 3.1. Participant Responses

Participants described encounters that occurred during an assortment of activities, including diving, snorkeling, boating, kayaking, swimming, with a variety of marine animals, such as turtles, seals, sharks, dolphins, fish, dugongs, seabirds, whales and rays and marine invertebrates (even though the research question enquired about experiences with marine mammals). Participants described watching wildlife “I look at the star fish and sea snails” and “a seal popped up just a few meters away” and also swimming with wildlife “surfing and swimming when dolphins, fish, turtles, sting-rays, and other marine life swam right past me”.

The researchers noted the participants’ use of similar language; common phrases and consistent expressions, revealing that nature interactions are innate to the human experience and even human consciousness [47] and they manifest in nature descriptions. Such similarities included multiple references to HAIs being “eye opening” and “humbling”, offering enhanced “insights “, “appreciation” and “awareness” and being “magical” or “magnificent”. Furthermore, there were widespread descriptions of the creation of connections and understandings, and how people felt “changed” by the experiences.

### 3.2. Losing the Nature Language?

Most participants were able to clearly articulate their experiences and connections with animals, at times even poetically, as this example shows, “The sea is a womb, nurturing, buoyant/Safe, warm, salty/Curious, crackling/I return again and again/To see and be seen” (*Recognising and being recognised by a non-human other, Love, belonging, connection*).

However, there were some respondents who appeared unable to elucidate their animal interactions, “it’s hard to find the words to adequately describe the amazing feeling that filled me when encountered with dolphins”. Another participant felt their descriptions were insufficient, “These words are so inadequate ... there really is something special that just can’t be put in words or pictures form”. Some participants actually stated they could not find the words to respond, for example, “I cannot put into words the experience. I do apologise” and “It’s really hard to explain how I am around mammals” and “It isn’t really explainable”.

Other participants provided simplistic positive responses such as “I liked it” and “A lot!!! Loved every part of it!” and “Always good to see animals in nature” or they paraphrased the question, such as, “If you watch wild animals OF Course it influences your understanding of nature. Dumb question” and “It’s neat to interact with wild critters”.

A further group of respondents, rather than describing their experiences, contrasted their real-life experience to virtual experiences on the screen or in a book, as a way of explanation, “That connection cannot be felt from a book or TV screen” and “It provided the reality a television screen could never supply” and “A deeper connection than I’d have by (for example) watching David Attenborough DVDs” and “They aren’t an illustration on a page anymore, they’re real, they’re out there, and I’ve seen them with my own eyes”.

The articulation difficulties that some participants encountered may fit with the previously discussed literature, regarding the loss of words to describe interactions—the loss of a nature language.

### 3.3. Themes: Psychological Descriptions and Interaction Patterns

The Psychological Descriptions from the encounters, whilst individual, concurred to common themes with four distinct Psychological Descriptions emerging; Positive emotions, Perspective gaining, Esteem and fulfilment, and Love, belonging, connection. The links between the IPs and PDs provided enlightening associations between participant experiences, Interaction Patterns and participant descriptions of mental health and wellbeing effects. Table 3 shows the number of responses to each of the five IPs and 4 PDs including crossovers. As can be seen, the most common Psychological Descriptions were *Perspective gaining* (253 references) and *Positive Emotions* (239 references). Some of the IPs did not correspond with the PDs, for example, the theme of *Encountering animals that can harm* did not elicit any descriptions of *Esteem, Fulfillment* or *Love, Belonging, Connection.* On the other hand, *Encountering, watching wildlife* had 70 co-occurring descriptions of *Positive Emotions.* Following on from Table 3, each Psychological Description theme is discussed in detail with representative quotes to allow the participant voices to be heard. 

#### 3.3.1. Positive Emotions

Positive feelings featured heavily in Psychological Descriptions of encounters. The theme *Positive emotions* was commonly co-coded with *Encountering/watching wildlife* and included accounts of relaxation and calmness, as one participant described, 

“Swimming with marine animals is also an extremely peaceful experience”. 

Another participant described a more intense happy reaction after swimming with dolphins, 

“Two days later and the feeling is still with me—a constant smile and occasional tear of gratitude and love” (*Encountering/watching wildlife, Positive emotions*)

Good feelings were also associated with the Interaction Pattern of *Recognising the signs of animals in nature,* for example, 

“Just like the seals, it’s just so awesome to be in their habitat sharing the same waves”, and, “it was awesome (in the literal sense) to be able to witness these fish in a feeding frenzy”.

#### 3.3.2. Esteem, Fulfillment 

Participants described being deeply or spiritually fulfilled by animal encounters (*Esteem, fulfillment*), especially when they felt acknowledged by an animal (*Recognising and being recognised by a non-human other*). A poignant example involves the participant who described an unexpected beach encounter with a single dolphin that helped them during a period of deep depression, 

“He helped me so much that day, I wasn’t alone it was as if he could see I was hurting in my soul and he helped the pain go away.” 

Using similar words, another participant described an unexpected sighting of a pod of dolphins (*Encountering/watching wildlife*) that helped her after a tough relationship break up, 

“the ocean was showing a bright side to my situation, as though it was rewarding me for being strong”. 

Further, descriptions of being in flow (*Esteem/fulfillment*) and being able to forget all else were also articulated, 

“I was able to forget about everything else going on around me while I was under the water watching the Dolphins” (*Encountering/watching wildlife*)

#### 3.3.3. Perspective Gaining

Participants described different insights or viewpoints gained from their interactions with marine animals. This theme highlights the potentiality for the HAI to provide scope for a shift in awareness to a less anthropocentric focus. An Indigenous Australian participant offered their unique perspective about marine animals, 

“I am Aboriginal and every living being is such an important part of our journey as humans. They have many stories to tell us. Our connection to other beings is very special spiritually”.

Some insights were personal, such as a participant’s interpretation of a message about how to cope with a difficult time, 

“I felt as though nature had given me a sign that it was time to open up. I felt very lucky for the earth to be looking out for me” (*Encountering/watching wildlife, Perspective gaining*)

New understandings sometimes involved realizing the power of animals, 

“To be in their environment, floating so the dolphins approached you when they wanted was a little intimidating and made you feel very small and insignificant and even vulnerable” (*Encountering animals that can harm, Perspective gaining*)

Other encounters with wildlife enabled participants to gain a broader, more ecocentric perspective, such as, 

“these experiences bring me a new look on sharks, realize the fragility of marine animals and make me aware on the fact that we are just a small part of the life on earth” (*Encountering/watching wildlife, Perspective gaining*)

Comments about evocative experiences triggering awareness also featured, 

“Looking into the eyes of a whale has been one of the most profoundly powerful experiences I have had in nature. It made me aware of the enormity of the marine world and the peaceful intelligence of these incredible animals” (*Recognising and being recognised by a non-human other, Perspective gaining*)

#### 3.3.4. Love, Belonging, Connection

Some participants described animal experiences that had forged or strengthened nature connections. Among the responses coded to this theme, HAIs were likened to exchanges with humans, 

“the fish communicated, we played hide and seek and chaseys. We were friends”, (*Recognising and being recognised by a non-human other, Love, belonging, connection*)

Subsequent effects on wellbeing from the experience of encountering wildlife were also articulated, 

“My mental and physical wellbeing have improved a lot—the term Dolphin Love Energy is how I describe the experience” (*Love, belonging, connection*)

Descriptions of shared understandings and mutual activity were also apparent in comments like these, 

“I felt like the dolphin understood that I had a disability”, and, “The connection with wild dolphins surfing the waves … in one of nature’s theatrical performances was extremely beautiful to be a part of” (*Interacting with the periodicity of nature, Love, belonging, connection*)

Projection of participants’ own love and belonging needs onto an animal, highlights a sense of mutuality and interrelatedness, 

“all living things strive for the same thing, to be safe and loved and connected to their own species”. 

Whilst some participant’s comments may be viewed as anthropocentric, it may also be suggested that they are attempting to connect their own understanding to sentient beings, such as wild animals, in valuing the importance of connection to kin.

Figure 1 presents a way to visually represent the interrelationships between wild animal experiences, the expression of Nature Interaction Patterns and the articulated Psychological Descriptions. The embedded circles show the connections between the study themes and experiences. The HAIs are rooted within the Interaction Patterns and the Psychological Descriptions flow from and to the Interaction Patterns.

Table 4 gives further examples of quotes that were coded to the combinations of IPs and PDs.

## 4. Discussion

The current study offered insight into common conceptualizations and articulations of the ways people interact with wild animals. The majority of participants were able to express how interactions with marine animals affected their subjective connectedness to nature. Nature Interaction Patterns [34], which concurred with Psychological Descriptions of wellbeing and mental health benefits were depicted. Rich narratives described meaningful participant accounts of myriad encounters with various animals in the marine environment. 

The study provided evidence for the proposition that when humans interact with animals in nature, they describe that their mental health and wellbeing is enhanced, supporting previous research on the topic [13,19,20,21,22,23,27]. In illustrating how marine animal interactions affected their nature connection, participants described psychological outcomes indicative of mental health benefits. Feelings of love, belonging, positive feelings, fulfillment and the gaining of perspective, were linked with the human–animal experience and the Interaction Patterns (Figure 1). A small group of participants were unable to find the words to describe their experiences, or the psychological outcomes, consistent with the concern of loss of the Nature Language [33].

*Positive emotions* in many forms featured in the stories that were told about animal encounters, from intense and powerful positive responses, to calmness and the discovery of inner peace. This was especially the case for participants who described the interaction patterns of *Encountering/watching wildlife, Reading the signs of animals,* and *Recognising and being recognised by a non-human other.* These thematic crossovers further validated that HAI wellbeing outcomes involve aspects of nature interaction and connection. The role of *Positive emotion* for wellbeing and flourishing has previously been explored [48] and in particular the role of wild animals in generating positive emotions and wellbeing [1,22,27]. 

Some participant stories about helpful feelings encompassed intimations of wider effects, through descriptions of spirituality, personal importance, esteem improvement and goal achievement. The Psychological Descriptions encompassed by the theme of *Esteem/fulfilment,* highlighted the intensity of meaning ascribed to the Interaction Pattern of *Encountering/watching wildlife*. Personal fulfilment is widely accepted in psychology as an important aspect of achieving the wellbeing need of self-actualisation [15].

It appeared from the narratives that the more positive and connected the participant felt towards animals, the greater the kinship with the natural world in general, as reflected in the themes of *Love, belonging, connection* and *Perspective gaining.* The sense of connection and bonding to animals, described in the narratives, may speak to an intrinsic sense of commonality and shared belonging to nature, sensed through mutual encounters [24]. This was particularly relevant in the overlaps between the Psychological Descriptions and the Nature Interaction Pattern of *Recognising and being recognised by a non-human other.* The felt connections of mutual recognition encompassed feelings of bonding, love and care which translated into a perceived sense of relationship, potentially serving to fill a facet of flourishing [48].

The gaining of perspective through animal experiences took many forms in the narratives, from changed internal viewpoints and putting one’s own suffering in perspective, to gaining knowledge of, and understanding the wider world. This enabled the individual to see their role in the larger ecosystem. Perhaps these psychological outcomes can contribute to improved mental health due to expansion of different and outwardly focussed viewpoints, rather than a narrow internal self-focus [5,13,49]. It is suggested that *Perspective gaining* could allow humans to see from the viewpoint of animals and nature, and to embrace a less anthropocentric view. Learning about, caring for and becoming advocates for nature, may in turn enhance the human benefits. Interactions with animals has been noted as beneficial for people who have had marine animal encounters [27]. As Yunkaporta [4] explains, the indigenous human view is that “the role of the custodial species is to sustain creation which is formed from complexity and connectedness”, (p. 273). Therefore, to be custodians of the earth, as the First Australians were, humans need to shift from an internal human focus, to an external, ecocentric holistic world view. This would allow an increase in awareness and understanding about animals and may encourage conservation and habitat protection. Further, it is possible that in order for humans to most fully benefit from HAIs, they also need to meet their reciprocal obligations to serve the relationship, in the same way that human-human relationships require reciprocity for true mutuality of benefit [11].

### Limitations and Future Directions

It is accepted that this study was designed to be qualitative and exploratory, thus the results are limited in their generalizability to other populations and their transferability to other wildlife or domesticated animals. It is further acknowledged that there can be many interpretations of a phenomena. There are various ways of translating data related to lived experience, and it will be influenced by the authors’ openness and relational stance [5]. Thus, it is acknowledged that the participant responses are subjective, and it is not known whether the participants are describing long-term or immediate effects and how long the described wellbeing benefits lasted for. This may be an area of subsequent investigation to examine whether effects are enduring. To increase generalizability and transferability of results, future research could investigate the phenomena via a longitudinal or quasi-experimental design with a larger, more varied sample in different contexts. It is further recognized that as the study is retrospective in nature and not experimental, and that the impacts on the animals are unknown as the interactions occurred prior to the data collection. While this study highlights the potential benefits to humans of HAIs, it is acknowledged that HAIs can and do have significant and negative effects on wildlife. While beyond the scope of this paper, negative effects of human -wildlife interactions are discussed elsewhere with regard to dolphin behavior [50], bird nesting [51] and turtle basking [52]. Therefore, the authors are not advocating or condoning people entering or pursuing interactions with wild animals.

## 5. Conclusions

Interactions with wild animals may enable gains in perspective via nature connectedness that encourage humans to reassess a constraining and restrictive anthropocentric focus. While the study does not explicitly gauge shifts in focus from anthropocentric to ecocentric views, it is suggested that enacting the Interaction Patterns associated with HAIs expands animal and nature perspectives. This may lead to increased reciprocal understandings and behaviors.

The study provides an initial exploration into the interpretation of HAIs through an Interaction Pattern approach incorporating nature connectedness. It is concluded that this approach is a valid way to interpret and to articulate HAIs. A way of expressing HAIs via a ‘nature language’ could help to re-instate the mutual and reciprocal connections that humans and animals have a history of enacting. Therefore, this study raises the suggestion that human wellbeing in HAIs cannot be fully realized without taking into account the effects on natural ecosystems and non-human beings involved. As Fisher [5] so aptly states; “for many of those who have indeed spent a life in open contact with nonhuman beings, the natural world is peopled with beautiful and mysterious others deserving of respect and solidarity” (p. 87).

Understanding and articulating Nature Interaction Patterns may reconnect humans with their innate voices and the external stories, both of which the modern world has encouraged alienation from. Further, such a conceptualization may encourage people to be open to think beyond themselves and their needs. This may rekindle an indigenous view of respectful and equal relationships with animals. It may be argued that reciprocity could assist in achieving mutual and therefore fuller and more beneficial relationships with nature and animals.

## Figures and Tables

**Figure 1 animals-11-00950-f001:**
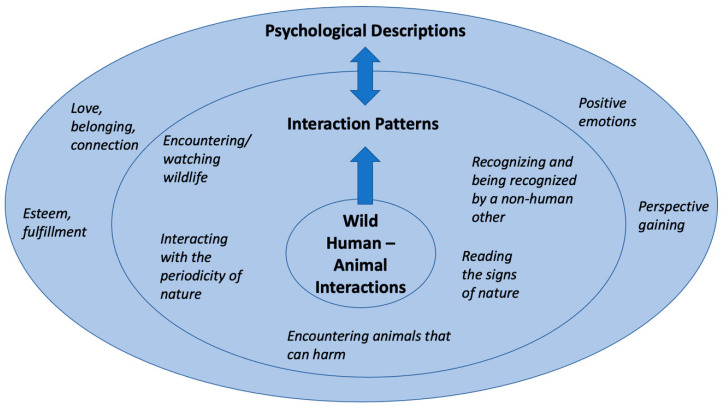
Visual representation of the embedded links between HAIs, Interaction Patterns and Psychological Descriptions.

**Table 1 animals-11-00950-t001:** Interaction Patterns and definitions.

Interaction Patterns (IP)	Definition
*Encountering animals that can harm*	Encountering potentially harmful or hurtful aspects of nature involving animals
*Encountering/watching wildlife*	Coming across wildlife in nature and observing/watching/swimming with them (only used if no other IP is relevant)
*Interacting with the periodicity of nature*	Encountering animals doing their natural activities, engaged in periodic cycles, changes or enacting patterns
*Reading the signs of nature*	Information by observing/witnessing a specific aspect of wild animals that provides relevant or meaningful information, such as learning about behaviours or habitats
*Recognizing & being recognized by a non-human other*	Interpreting that there has been a sensory connection or acknowledgement with an animal

**Table 2 animals-11-00950-t002:** Psychological Descriptions and definitions.

Psychological Description (PD)	Definition
*Positive emotions*	Reporting of positive affect, feelings, emotions or impacts. (happy, calm, excited, awed, appreciative, wonder, respectful etc)
*Esteem, fulfillment*	Improvement in one’s sense of self with regard to self-esteem or feeling fulfilled. gaining a sense of satisfaction or accomplishment
*Perspective gaining*	Gaining a different point of view or realisation or inspiration. Gaining a different way of looking at something
*Love, belonging, connection*	Feeling loved, cared for, included or connected. Sense of belonging or fitting in—in the world, in nature, etc.

**Table 3 animals-11-00950-t003:** Numbers of responses coded to all IP and PD themes.

	Encountering Animals That Can Harm	Encountering/Watching Wildlife	Interaction With the Periodicity of Nature	Reading the Signs of Nature	Recognising and Being Recognised by a Non-Human	Esteem, Fulfillment	Love, Belonging, Connection	Perspective Gaining	Positive Emotions
Encountering animals that can harm	17								
Encountering/watching wildlife	0	247							
Interaction with the periodicity of nature	0	2	35						
Reading the signs of nature	0	14	1	120					
Recognising and being recognised by a non-human other	0	1	4	2	97				
Esteem, fulfillment	0	29	6	1	9	102			
Love, belonging, connection	0	15	4	3	6	3	122		
Perspective gaining	1	18	2	15	4	4	5	253	
Positive emotions	1	70	8	20	16	10	4	8	239

**Table 4 animals-11-00950-t004:** Overlaps between IPS and PDs.

	Esteem, Fulfillment	Love, Belonging, Connection	Perspective Gaining	Positive Emotions
Encountering animals that can harm	-	-	“To be in their environment, floating so the dolphins approached you when they wanted was a little intimidating and made you feel very small and insignificant and even vulnerable”.	“I can enjoy the activities I take part in and not being terrified”
Encountering/watching wildlife	“surfing with dolphins—its what its all about. The spirituality of surfing is profound”.	“the fish communicated, we played hide and seek and chaseys. We were friends.”	“to see whales or dolphins in their own environment you realise that we have to protect the oceans so that they can have the best life possible”	“Swimming with marine animals is also an extremely peaceful experience”.
Interaction with the periodicity of nature	“Realised how lucky I was to live next to a body of water that whales visited. Was a very special moment.”	“Surfing with dolphins around makes me feel like we are connected in some way even if its in the simple act of using a wave for transport”	“Watching whale sharks and their understanding of how and why they were travelling gave me greater perspective”	“To see dolphins in their natural environment and enjoying riding waves is a beautiful thing. It makes me happy to see dolphins while I am surfing”
Reading the signs of nature	“When the dolphins approached, I swam under water and I could hear them calling … The only way to explain this is that the experience was magical”	“observing wild animals in their natural environment inspires me, affirms my love for nature”	“Seeing whales and dolphins in their natural environment made me feel connected to nature”	“it was awesome (in the literal sense) to be able to witness these fish in a feeding frenzy”
Recognising and being recognised by a non-human other	“marine animals like dolphins swimming and engaging with you freely, it is like a spiritual connection. In that you feel like they can understand you and you can understand them. Magical”	“creating a bond between the dolphin and myself”.	“Looking into the eye of a whale helps you understand its intelligence and inquisitiveness”	“They usually come and have a look, sometimes spy hop, mostly swim under and roll to their sides to check me out. Its always a highlight”

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
