# Peer review of "Human–Animal Interactions: Expressions of Wellbeing through a “Nature Language”"

_animals, 2021, doi:10.3390/ani11040950_

Round 1

Reviewer 1 Report

Please see the attached doc.

Author Response

Dear Reviewer,

Thank you for your time in reviewing our paper. We have attended to your concerns and suggestions and made changes on the manuscript and in the attached document.

Thank you

Rachel Yerbury and Samantha Lukey 

Reviewer 2 Report

This is an interesting and original manuscript that touches on some critically important topics, including wellbeing and human-animal interactions. Overall, this manuscript is very well-written and thorough. I appreciate that this was qualitative and exploratory study, but I do think some minor changes could increase the study's external validity. Below are some specific comments/questions that I think would be beneficial to address in a revised version:

  1. Section 1.1 is interesting, yet it is unclear to me that it directly applies to the present study. Perhaps the information contained in this section could be combined with another section? Beginning the discussion with HAI's seems more relevant.
  2. The manuscript would be improved with some quantitative data. For example, what percentage of responses were included in a particular category? Additionally, these results could be clearly displayed in a table for ease of interpretation.
  3. It's clear that you took steps to ensure reliability. Did you conduct any reliability testing that could be reported in the manuscript?
  4. Overall, the results are very detailed, but challenging to follow. It seems like some of the information contained in this section may be more relevant to the discussion. The discussion section would be more robust as a result.

Author Response

(The authors gave the same response as above.)

Round 2

Reviewer 1 Report

Thank you very much for addressing all of my comments. I feel reassured that the potential impacts on animals have now been flagged up in the paper and the needs and perceptions of humans and non-humans have been better balanced.

I just have one outstanding question - line 49 refers to 'mutually beneficial hunting'. Please could you explain what this is - does it mean cooperative hunting by humans and predators of a different species? Otherwise, how does it benefit the hunted animal, unless this is with respect to the overall fitness of the population? A genuine question, from someone outside the field.

Author Response

Dear Reviewer,

Thank you for your questions. This is a very interesting topic!! See my responses below, did you also wish me to put some extra  text into the manuscript about this?

There are many examples of humans and cetaceans hunting together for mutual benefit. In Australia in Eden, NSW, Aboriginals, then white settlers were assisted by killer whale (orca) to heard in Southern Right whales humpbacks. The orca were then given the tongue as payment. There is a good book on this called "The Killers of Eden" by Tom Meade.

There is also co-operative fishing in Laguna Brazil with dolphins herding the fish into the shore towards the fishermen with nets. The dolphins also find this helpful to catch the fish as the fishermen create a wall. There are many articles and youtube videos on this.  

Here is a link with more examples.  http://eyes4earth.org/2012/11/dolphin-assisted-fishing/

Thanks for your interest

Rachel